# Construction of Cellular Substructure in Laser Powder Bed Fusion

**Yafei Wang [1], Chenglu Zhang [1], Chenfan Yu [1], Leilei Xing [1], Kailun Li [1], Jinhan Chen [1], Jing Ma [1], Wei Liu [1,*] and Zhijian Shen [1,2,*]**

[1]  State Key Laboratory of New Ceramic and Fine Processing, School of Materials Science and Engineering, Tsinghua University, Beijing 100084, China; wyf15@mails.tsinghua.edu.cn (Y.W.); chocneco@gmail.com (C.Z.); ycf15@mails.tsinghua.edu.cn (C.Y.); xingll16@mails.tsinghua.edu.cn (L.X.); likl16@mails.tsinghua.edu.cn (K.L.); chenjinh17@mails.tsinghua.edu.cn (J.C.); ma-jing@mail.tsinghua.edu.cn (J.M.)

[2]  Arrhenius Laboratory, Department of Materials and Environment Chemistry, Stockholm University, S-106 91 Stockholm, Sweden

*  Correspondence: liuw@tsinghua.edu.cn (W.L.); shenzhijian@tsinghua.edu.cn (Z.S.); Tel.: +86-010-62772852 (W.L.); +86-010-62788237 (Z.S.)

**Abstract:** Cellular substructure has been widely observed in the sample fabricated by laser powder bed fusion, while its growth direction and the crystallographic orientation have seldom been studied. This research tries to build a general model to construct the substructure from its two-dimensional morphology. All the three Bunge Euler angles to specify a unique growth direction are determined, and the crystallographic orientation corresponding to the growth direction is also obtained. Based on the crystallographic orientation, the substructure in the single track of austenitic stainless steel 316L is distinguished between the cell-like dendrite and the cell. It is found that, with the increase of scanning velocity, the substructure transits from cell-like dendrite to cell. When the power is 200 W, the critical growth rate of the transition in the single track can be around 0.31 ms$^{-1}$.

**Keywords:** laser powder bed fusion; substructure; model; growth direction; crystallographic orientation; cell; cell-like dendrite

## 1. Introduction

Laser powder bed fusion (LPBF) is a metal additive manufacturing technology. LPBF was developed to directly fabricate metal parts and tools with complex shape [1]. Later, it was found that metal parts fabricated by LPBF had a hierarchical structure [2]. Of particular interest is the solidification substructures observed in several LPBF alloys [3]. The substructure appeared to be cells or corrugations in the etched two-dimensional (2D) section, and was believed to exist everywhere in the bulk sample [4–6]. Some researchers realized that the substructure had a three-dimensional (3D) morphology [7,8], which was in the form of a prismatic array [9]. Nevertheless, the 3D morphology of the substructure has seldom been validated [10].

The general method to study the substructure can be based on the etched 2D morphology [2–8,10,11], which is cheap, easy to operate and effective. However, the 3D information on the growth direction of the substructure cannot be obtained directly from the 2D morphology. Chen et al. built a model of hexagonal prism to analyze the angle between the growth direction and the normal direction of the 2D section [7]. Arısoy et al. studied the angle of the growth direction to the building direction [8].

These studies only took one angle into account [7,8], while three angles are necessary to specify a unique direction in 3D [12]. Hitherto, there is a lack of research to construct the substructure from the 2D morphology in consideration of three angles.

Kurz indicated that the cellular substructure could be a cell or a cell-like dendrite [13]. The cell grew along the direction of the thermal gradient, and the dendrite grew with one of its preferred crystallographic orientations, while the cell-like dendrite could grow in a direction between the thermal gradient and the preferred crystallographic orientation. However, it is difficult to distinguish the cell and the cell-like dendrite from the morphology, since the cell-like dendrite might slightly branch [14,15]. It is possible to make the distinction from the crystallographic orientation of the substructure, and Sun et al. obtained the crystallographic orientation corresponding to the growth direction by transmission electron microscopy (TEM) [11]. However, the location information is usually lost in TEM, which can be further correlated with the local solidification parameters [16]. Herein, it is attractive to quantify the crystallographic orientation corresponding to the growth direction from the etched 2D morphology, which enables the distinction between the cell and cell-like dendrite, and the correlation with the local solidification parameters.

This research tried to construct the structure of the substructure in austenitic stainless steel (SS316L) fabricated by LPBF. A model was built to construct the substructure from the 2D etched morphology. Three angles to specify a unique growth direction of the substructure were tended to be determined by the combination of the geometrical model and the local thermal condition. Two tests were designed to validate the growth direction. Furthermore, the crystallographic orientation corresponding to the growth direction was obtained with the help of local orientation data. Yadroitsev indicated that a part is manufactured track by track with LPBF technology, and that the properties of the part depended strongly on the properties of each single track and each single layer [17–19]. The study on the morphology and microstructure of the single track facilitates a comprehensive scientific understanding of the LPBF process-related phenomena and the effect of process parameters. In this research, the model was applied to construct the substructure from the single tracks at different scanning velocities. Based on the crystallographic orientation, the cell and cell-like dendrite were distinguished. The transition between the cell and the cell-like dendrite was compared to previous experimental and theoretical research.

## 2. Materials and Methods

An AM 250 LPBF system (Renishaw, New Mill, UK) equipped with an Nd:YAG fiber laser (200 W) was used, which focused at a spot with a diameter of about 75 μm. The diameter of the SS316L powder (Renishaw, New Mill, UK) ranged from 15 μm to 45 μm. An SS316L sheet with a thickness of 3 mm was used as the substrate. Table 1 discloses the nominal chemical composition of the SS316L powder and the substrate (Renishaw, New Mill, UK). The as-received sheet was heat treated as follows: First preheating to 850 °C in 2 h and holding for 1 h, then heating to 1000 °C in 1 h and holding for 10 h, finally cooling with the furnace. Yadroitsev indicated that the continuous and stable single track is essential to fabricate a 3D part with high quality [17–19]. The stability of the single track can mainly depend on the parameters of power, scanning velocity and powder thickness [18], and the scanning velocity seems to be a more flexible and suitable variable to obtain the desirable track microstructure and geometry [17]. In this research, the power and powder thickness are fixed, and the scanning velocity is varied. The laser power and the exposure time of each spot were set to be 200 W and 80 μs, respectively. When fabricating the single track, the point distance was set to be 25 μm, 50 μm and 100 μm, which corresponded to a scanning velocity of 0.31, 0.63 and 1.25 ms$^{-1}$, respectively. The hatch distance was set to be 800 μm, and the thickness of the metal powder layer was 50 μm. A cubic sample was fabricated with the point distance of 50 μm, the hatch distance of 60 μm, the layer thickness of 30 μm and the zigzag scanning strategy. The samples were mounted and polished (9 and 3 μm diamond suspension, and 0.05 μm colloidal silica suspension). The as-polished samples were finally polished by a 0.06 μm colloidal silica suspension using Vibromet 2 for 1 h. To show the structure of the substructure, the section normal to the scanning direction from the cubic sample was electrolytically etched at 6 V for 40 s using a 10% oxalic acid aqueous solution.

**Table 1.** Chemical composition of SS316L powder and the substrate.

| Element | Fe | Cr | Ni | Mo | Mn | Si | N | O | P | C | S |
|---------|------|------|------|------|------|------|------|------|------|------|------|
| wt pct | balance | 16 to 18 | 10 to 14 | 2 to 3 | 2 | 1 | 0.1 | 0.1 | 0.045 | 0.03 | 0.03 |

To show the outline of the substructure, the transverse cross section from the single track was etched for 30–40 s by a solution with $HCl:HNO_3:H_2O$ = 1:1:1. The morphology of the substructure was characterized by scanning electron microscope (SEM, TESCAN, Brno, Czech Republic), and the orientation map was collected by electron backscattered diffraction (EBSD, Oxford Instrument, Oxford, UK) at a step size of 0.5 μm.

## 3. Results and Discussion

### 3.1. 3D Morphology of the Substructure

Figure 1 shows the substructure from the cubic sample. As seen in Figure 1a, the 2D section is divided into some irregular colonies, which are marked out by the black dotted line. Inside each colony, the substructure is similar in morphology. The boundary of the colony may correspond to the large angle grain boundary [20]. As seen in Figure 1b–d, the substructures are oriented arrays in 3D, and they incline at different angles to the section. In Figure 1b–d, the substructure is inclined, parallel and perpendicular to the section, respectively. The substructure in colony (c) indicates that the substructure is straight, parallel to each other and across the colony in 3D. Despite the different morphology, the substructure exists everywhere in the section. In consideration of its 3D morphology, the substructure can exist everywhere in the cubic sample.

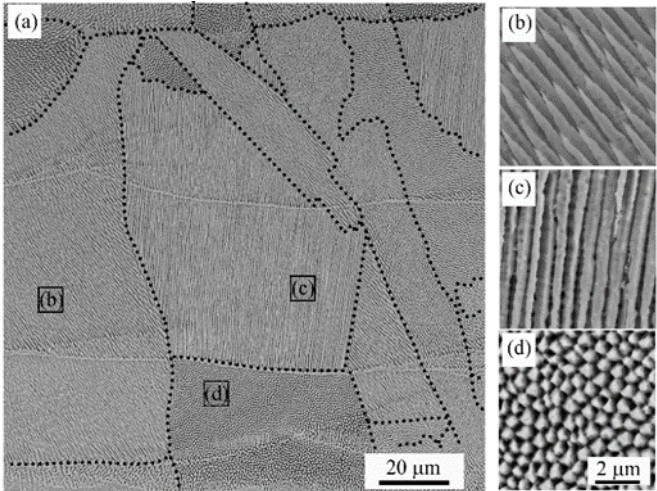

**Figure 1.** The substructure in the cubic sample. (**a**) Low magnification. (**b–d**) High magnification. The boundaries of the colonies are marked out by the black dotted line. (**b**) The substructure is inclined to the section. (**c**) The substructure is parallel the section. (**d**) The substructure is perpendicular to the section.

### 3.2. Model of the Substructure

The substructure array can be described by a bundle of hexagonal prisms [21], and the model of a single hexagonal prism is shown in Figure 2d. As seen in Figure 1, hexagons in the 2D section vary in morphology due to different inclination angles to the section. Hence, the key to calculate the growth direction is to build the function of morphology with regard to the inclination angle, which is shown as follows.

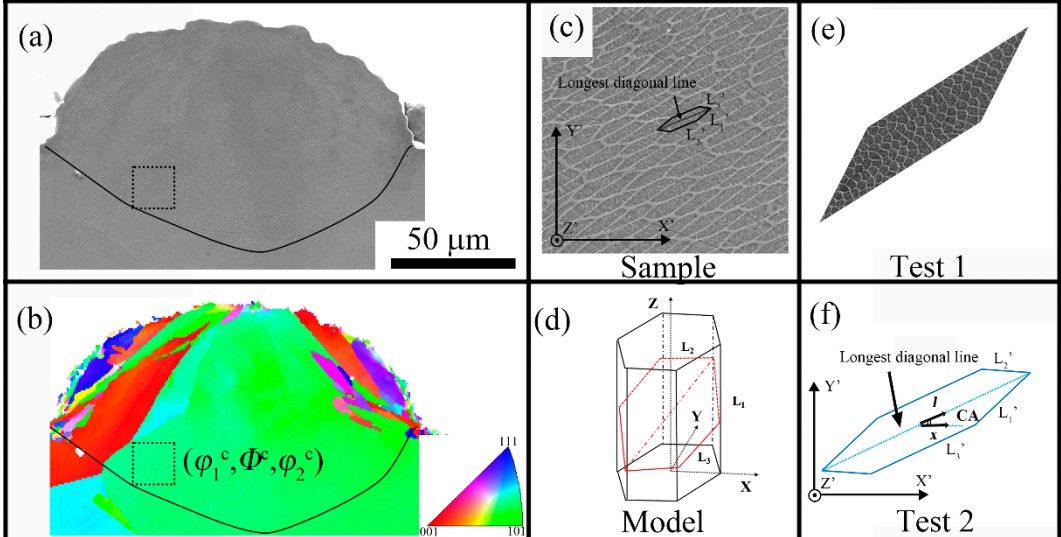

**Figure 2.** The process to calculate the growth direction of the substructure in the single track. (**a**) is the etched transverse cross section and (**b**) is the EBSD map corresponding to the section. The boundary of the track is marked out by the black dot line. (**c**) is the region sampled from the transverse cross section, and the outline of the substructure is light gray. A real hexagon is selected from this region, whose boundary is marked out by black line. (**d**) is the model of a hexagonal prism with the top surface of a regular hexagon. (**e**) is the projected figure of the sampled region. (**f**) is the transformative hexagon sectioned from the hexagonal prism according to the growth direction.

The hexagon prism in Figure 2d has the top face of a regular hexagon, whose sides equal to 1. A coordinate system (XYZ), $\{K_P\}$, is built on the prism. Three Bunge Euler angles ($\varphi_1$, $\Phi$, $\varphi_2$) can define a rotation from $\{K_P\}$ to a sample coordinate system (X'Y'Z'), $\{K_S\}$ [12]. As seen in Figure 2d, when intercepting the prism with a section of (X'OY'), a red hexagon appears in this section. The equation representing the section of (X'OY') in $\{K_P\}$ is:

$$\sin\varphi_1\cdot\sin\Phi\cdot x - \cos\varphi_1\cdot\sin\Phi\cdot y + \cos\Phi\cdot z = 0. \tag{1}$$

According to linear algebra, the opposite sides of the red hexagon are parallel and equal in length. The lengths of three adjacent sides can be expressed as:

$$L_1 = 1 + (p + q)^2 \tag{2a}$$

$$L_2 = 1 + 4p^2 \tag{2b}$$

$$L_3 = 1 + (p - q)^2 \tag{2c}$$

where

$$p = 1/2\cdot\sin\varphi_1\cdot\tan\Phi \tag{2d}$$

$$q = \sqrt{3}/2\cdot\cos\varphi_1\cdot\tan\Phi \tag{2e}$$

The above equations indicate that the lengths of the sides are functions of ($\varphi_1$, $\Phi$).

On the other hand, a real hexagon can be selected from the etched section, as shown in Figure 2c. Its sides are measured as ($L_{1'}$, $L_{2'}$, $L_{3'}$). By recombining Equations (2a)–(2e), the values of $p$ and $q$ are obtained by solving the following equations:

$$p^2 + q^2 + 2pq + 1 - m(1 + 4p^2) = 0 \tag{3a}$$

$$p^2 + q^2 - 2pq + 1 - n(1 + 4p^2) = 0 \tag{3b}$$

where

$$m = (L_{1'}/L_{2'})^2 \tag{3c}$$

$$n = (L_{3'}/L_{2'})^2 \tag{3d}$$

The ratio of $p$ and $q$ is:

$$p/q = \tan(\varphi_1)/\sqrt{3} \tag{4}$$

In consideration of the six-fold symmetry of the hexagon prism, $\varphi_1$ is in the range of $[0, \frac{\pi}{3})$. Given a real hexagon, Equation (4) will return one solution to $\varphi_1$.

The value of $\tan\Phi$ can be calculated from Equation (2d) or (2e). Given the value of $\tan\Phi$, there are four possible $\Phi$:

$$\Phi(1) = \arctan(2q/\sin\varphi_1) \tag{5a}$$

$$\Phi(2) = \Phi(1) + \pi \tag{5b}$$

$$\Phi(3) = -\Phi(1) \tag{5c}$$

$$\Phi(4) = -\Phi(1) + \pi \tag{5d}$$

The calculation of $\varphi_2$ relies on the cardinal direction of the hexagon. On the one hand, the cardinal direction of a real hexagon is measured as the angle between the longest diagonal line and X'-axis of the section, i.e., in $\{K_S\}$. The cosine value of the angle is labeled as CA. On the other hand, the direction of X'-axis of $\{K_S\}$ can be expressed as follows in $\{K_P\}$:

$$x = (\cos\varphi_1 \cdot \cos\varphi_2 - \sin\varphi_1 \cdot \sin\varphi_2 \cdot \cos\Phi, \sin\varphi_1 \cdot \cos\varphi_2 + \cos\varphi_1 \cdot \sin\varphi_2 \cdot \cos\Phi, \sin\varphi_2 \cdot \sin\Phi) \tag{6}$$

The direction of the longest diagonal line can be:

$$l = (-1, \sqrt{3}, (\sin\varphi_1 + \sqrt{3}\cos\varphi_1)\tan\Phi) \tag{7}$$

which is marked out as the red dot line in Figure 2d. The measured cardinal direction is the angle between $x$ and $l$:

$$x \cdot l / \|l\| = CA \tag{8a}$$

$$(\sin\varphi_2)^2 + (\cos\varphi_2)^2 - 1 = 0 \tag{8b}$$

Given one $\Phi$, Equations (8a) and (8b) will return two possible $\varphi_2$.

The above calculations can only return the possible mathematical solutions to $\Phi$ and $\varphi_2$, but not the unique solution. It is likely to determine the unique solution in consideration of the local thermal condition of the substructure. The growth direction of the prism in $\{K_S\}$ is the direction of Z-axis of $\{K_P\}$, which is the third column of the rotation matrix in Equation (11):

$$v = (\sin\varphi_2 \cdot \sin\Phi, \cos\varphi_2 \cdot \sin\Phi, \cos\Phi) \tag{9}$$

Due to the thermal condition in the single track, the cellular substructure can grow along the scanning direction and towards the center and the top of the track [7,14]. The location (x', y', z' = 0) of the cellular substructure can be read from the 2D section. The signs of the components of $v$ are checked to see whether they agree with the local thermal condition. The sign of the third component, $\cos\Phi$, can be determined from the scanning direction. Hence, the possible $\Phi$ in Equations (5a–d) are reduced from four to two. For each $\Phi$, two possible $\varphi_2$ are obtained from Equation (8a) and (8b), and the possible vs. are calculated by Equation (9). The second component of $v$, $\cos\varphi_2 \bullet \sin\Phi$, must be positive to ensure the upward growth of the cellular substructure. The sign of the third component of $v$ can be opposite to the sign of x' due to heat conduction. It means that if the substructure locates on the left side, it shall grow towards the right side. Herein, the unique $\Phi$, $\varphi_2$, and $v$ can be selected from the possible mathematical solutions.

Two tests are designed to validate the unique solution. One is to project Figure 2c onto the plane normal to the growth direction, as shown in Figure 2e. The outline of the substructure in the projected figure shall be as close as possible to the regular hexagon. The unique solution may be slightly tailored to better satisfy the tests. The other is to draw a hexagon by intercepting the prism growing along the growth direction with the section of (X′OY′), as shown in Figure 2f. The morphology and cardinal direction of the hexagon shall be as similar as possible to that of the substructure in Figure 2c.

Furthermore, the crystallographic orientation, $(\varphi_1^C, \Phi^C, \varphi_2^C)$, corresponding to the same area, are read from the EBSD map, as seen in Figure 2b. The crystallographic orientation of $v$ is:

$$cv = g \cdot v \tag{10}$$

where $g$ is the rotation matrix, which rotates the sample system, $\{K_S\}$, to the crystal system, $\{K_C\}$. The rotation matrix is calculated as follows [12]:

$$g = \begin{bmatrix} \cos\varphi_1^C \cdot \cos\varphi_2^C - \sin\varphi_1^C \cdot \sin\varphi_2^C \cdot \cos\Phi^C, & \sin\varphi_1^C \cdot \cos\varphi_2^C \cos\varphi_1^C \cdot \sin\varphi_2^C \cdot \cos\Phi^C, & \sin\varphi_2^C \cdot \sin\Phi^C \\ \cos\varphi_1^C \cdot \sin\varphi_1^C - \sin\varphi_1^C \cdot \cos\varphi_1^C \cdot \cos\Phi^C, & -\sin\varphi_1^C \cdot \sin\varphi_1^C + \cos\varphi_1^C \cdot \cos\varphi_1^C \cdot \cos\Phi^C, & \cos\varphi_2^C \cdot \sin\Phi^C \\ \sin\varphi_1^C \cdot \sin\Phi^C & -\cos\varphi_1^C \cdot \sin\Phi^C & \cos\Phi^C \end{bmatrix} \tag{11}$$

This model can be a general way to construct the solidification microstructure from the 2D section, since it depends on the quantitative data from the etched 2D morphology. Three Bunge Euler angles to specify a growth direction can all be determined. The unique solution can be selected in consideration of the local thermal condition. Hence, this model can offer more information on the growth direction compared to the previous research [7,8], which took only one angle into account.

Moreover, image processing techniques may increase efficiency when processing a large number of images [8]. The image processing techniques may also lower the error of the calculation of the growth direction, because the tests are done by eyes in the present case.

Because all the three Bunge Euler angles of $v$ have been determined, the crystallographic orientation corresponding to the growth direction can be easily obtained with the EBSD technique. The location information of the selected zone can be quickly read from the etched section, which can be further related to the local solidification parameters [16]. This model can also be more efficient than TEM in obtaining a large number of crystallographic orientations.

### 3.3. Application of the Model

Because the local thermal condition can be easily determined in a single track, the model is applied to the transverse cross section from the single track. A set of growth directions and their crystallographic orientations were obtained from different colonies in the sections from the single tracks at different scanning velocities. They are plotted in the stereographic pole figure (PF) and the stereographic inverse pole figure (IPF), respectively. The images from the scanning electron microscope have been processed by the technique of gradation analysis to better show the outline of the substructure.

Figure 3a–c are the PFs of the growth directions, where a vector from the center point to the data point represents the growth direction. As can be seen, all of the substructure grows upward and toward the center of the track. Such growth tendency results from the thermal gradient in the track [7,14]. In each PF, the growth directions locate around an arc, which can partly result from the arc-like melting boundary, as seen in Figure 2a. With the increase of scanning velocity, the arc gets closer to the building direction (BD). The inclination angle between the growth direction and BD is calculated, whose average and standard deviation are listed in the PFs. The average inclination angle decreases from 44.91° to 36.91° as the scanning velocity increases, which means that the substructure grows more vertically at the faster scanning velocity. The standard deviation of the inclination is about 13°~16°, which is relatively large compared to the average inclination angle. The variation may result from the transient thermal gradient and the random crystallographic orientation of the polycrystalline substrate.

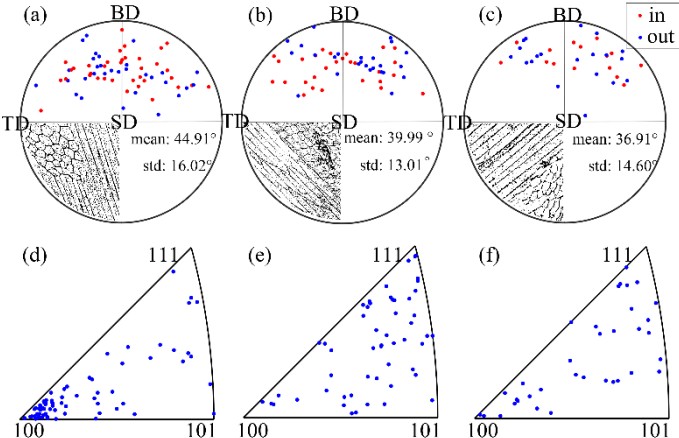

**Figure 3.** The stereographic pole figures of the growth direction and the stereographic inverse pole figures of the corresponding crystallographic orientation. Both (**a**) and (**d**) are calculated from the substructure in the track at scanning velocity of 0.31 ms$^{-1}$. Both (**b**) and (**d**) are at the scanning velocity of 0.63 ms$^{-1}$. Both of (**c**) and (**e**) are at the scanning velocity of 1.25 ms$^{-1}$. In the pole figures, the inward growth directions are plotted as red points, while the outward growth directions are plotted as blue points.

The crystallographic orientation of the growth direction is shown in Figure 3d–f. Only the substructure in the track at scanning velocity of 0.31 ms$^{-1}$ tends to grow with the <001> crystallographic orientation. In the other two tracks, the crystallographic orientation of the substructure is rather random, and no obvious tendency to be around <110> or <111>. As seen in the subplot in the PFs, the morphologies of the substructure do not show any branching, and they are cell-like.

In addition, the size of the substructure does not vary steeply with the increase of scanning velocity, which is also observed by other researchers [6,7,20]. Nevertheless, the crystallographic orientation indicates that a transition in solidification mode occurs with the increase of scanning velocity. The transition is whether the crystallographic orientation of the growth direction is along <001> or not. As shown in Figure. 1, the cellular substructure is straight in three dimensions. It is believed that the selection of growth orientation, i.e., the solidification mode, can start from the very beginning of solidification. According to the definition from Kurz [13], the dendrite grew with one of its preferred crystallographic orientations, and the cell grew along the direction of the thermal gradient. For alloys with a cubic crystal system, the <001> crystallographic orientation is found to be the preferred orientation [7,11,13]. At a low scanning velocity of 0.31 ms$^{-1}$, the substructure in the track tends to grow along with the <001> crystallographic orientation, which is more likely to be cell-like dendrite. At a higher scanning velocity, there can be no preferred crystallographic orientation, and the substructure is more likely to be that of a cell. On the one hand, the direction of the thermal gradient at different locations of the melting pool is different [6,7,22]. On the other hand, a polycrystalline substrate is used. Given the direction of the thermal gradient at a fixed location in the melting pool, the crystallographic orientation corresponding to the thermal gradient can vary as the melting pool moves [22]. Since the cell grows along the direction of the thermal gradient, the corresponding crystallographic orientation can be random.

The transition between cell and dendrite can occur in the low and high growth rate regime, which corresponds to the constitutional undercooling limit and the absolute stability limit, respectively [23]. In the low growth rate regime, it was found that with the increase of growth rate, the cell transited to the dendrite, and the cell-like dendrite tended to grow along the <001> crystallographic orientation [24,25]. In the high growth rate regime, the tendency is reversed [14,15,26]. In the present case, the cell-like dendrite transits to the cell with the increase of scanning velocity, which can belong to the transition in the high growth rate regime. Such a transition at a high growth rate agrees with the experimental observation [14,15,26] and with the theoretical prediction [23]. This indicates that the solidified

conditions in LPBF can be close to absolute stability. In the track at a scanning velocity of 0.31 ms$^{-1}$, there are also some points far away from the <001> crystallographic orientation, which may be cells. Besides, all the substructure in the other two tracks at the higher scanning velocity is cell. It is estimated that the critical growth rate of the transition may be around 0.31 ms$^{-1}$ in the single track with power of 200 W. This critical growth rate is also comparable in magnitude with the experimental and theoretical results [14,23]. It is noted that the critical growth rate in the present case is comparable to that obtained from the commonly used parameters in LPBF [2,4–8,11,27,28], and in other rapid solidification processes [14–16,23]. In these cases, it is suggested to be cautious to assume that the cellular substructure grows along the <001> crystallographic orientation. Besides, the development of the substructure may also depend on the as-solidified substructure in LPBF [7]. Further research may be achieved with the model presented in this research.

## 4. Conclusions

The substructure in SS316L fabricated by LPBF can grow in the form of an oriented array in 3D. For such substructure, a model is built to calculate its growth direction from the 2D morphology. The three Bunge Euler angles to a unique growth direction is determined in consideration of local thermal condition. Furthermore, the crystallographic orientation corresponding to the growth direction is obtained. The model has been successfully applied to construct the substructure in the single tracks fabricated at different scanning velocities. When the scanning velocity ranges between 0.31 ms$^{-1}$ and 1.25 ms$^{-1}$, the growth direction of the substructure is random, and it tends to be along BD with the increase of scanning speed. The transition from cell-like dendrite to the cell is identified by the crystallographic orientation. At low scanning velocity, the substructure is cell-like dendrite growing along with the <001> crystallographic orientation. At high scanning velocity, the substructure is cell, whose crystallographic orientation is also random. The critical growth rate of the transition in the single track with power of 200 W can be around 0.31 ms$^{-1}$.

**Author Contributions:** Conceptualization, Y.W.; methodology, Y.W., C.Z.; validation, Y.W., C.Y. and L.X.; formal analysis, Y.W., K.L., J.C.; investigation, Y.W.; resources, Y.W., L.X.; data curation, Y.W., C.Z.; writing—original draft preparation, Y.W.; writing—review and editing, J.M., W.L., Z.S.; visualization, Y.W.; supervision, W.L., Z.S.; project administration, Z.S.; funding acquisition, Z.S., W.L.

**Funding:** This research was funded by the Joint Funds of the National Natural Science Foundation of China, grant number U1605243.

**Acknowledgments:** The authors acknowledge Xin Zhou and Dianzhen Wang for the discussion of the model, and acknowledge Yuanyi Duan for the revision of the manuscript.

**Conflicts of Interest:** The authors declare no conflict of interest.

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
