# Peer review of "Construction of Cellular Substructure in Laser Powder Bed Fusion"

_metals, doi:10.3390/met9111231_

Round 1

Reviewer 1 Report

This paper aims to build a model to construct the 3D structure from 2D morphology of materials fabricated by selective laser melting. The way to calculate the growth direction from the 2D morphology is reasonable, and its usefulness has been confirmed.

Overall, I think this paper should be published in Metals.

I have tiny comments/question as below.

Figure caption of Figure 3: Is the scanning velocity of 63 ms-1 for Fig. 3(b) and (d) correct? I think it was 0.63ms-1 , because the authors explained “With the increase of scanning velocity, the arc gets closer to building direction (BD)”.

Could you explain more why the growth orientation of cell-like dendrite tends to be <001> and that of cellular becomes random? When we consider the interface instability at lower growth velocity during the directional solidification, a flat crystal/melt interface changes from flat to cellular to cellular dendrite in case that the growth velocity gradually increase during directional solidification. In that case, growth direction of a grain does not change when the growth morphology changes from cellular to dendrite. On the other hand, in this study, the phenomenon was observed at much higher growth velocity, as authors mentioned. In such at higher growth velocities, isn’t the grain size (the number of grain) related to the grain orientation distribution?

Reviewer 2 Report

There are several issues that should be addressed in the manuscript before further consideration for publication.

1. Suggest the authors to use ISO/ASTM terminology when describing the process

Lee et al. (2018),3D bioprinting processes: A perspective on classification and terminology, International Journal of Bioprinting 4(2), 151

2. Why are single tracks used? How representative are they for the SLM process where there is cyclic thermal phenomena?

3. It is understood that process parameters have effect on the microstructure. How are the parameters used determined? How can the results obtained be applied for other process parameters set? For other materials?

Khorasani et al. (2019), A comprehensive study on variability of relative density in selective laser melting of Ti-6Al-4V, Virtual and Physical Prototyping 14(4), 349-359 Sing et al. (2018), Selective laser melting of titanium alloy with 50 wt% tantalum: Effect of laser process parameters on part quality, International Journal of Refractory Metals and Hard Materials 77, 120-127

4. Any validation done for the model?

Author Response

Please see the attachment。

Reviewer 3 Report

The paper is well structured. Its content is consistent with the title and abstract. The conclusions are in accordance with the objectives stated by the authors.

The paper needs an extensive revision of the English language. For example:

Line 28: The formulation "is one kind of metal additive manufacturing techniques" should be replaced by "is a metal additive manufacturing technology"

Lines 34 and 35: The formulation "3D structure of the substructure" should be replaced by something that avoids the repetition, for example "3D morphology of the substructure". The formulation "3D structure of the substructure" is frequently used in the paper.

Line 75 - the word "facility" should be replaced by "system"

Lines 85, 89 and 91 - the word "about" should be removed.

Lines 81, 82 and 254 - The term "power" should be replaced by "laser power"

Line 121: The formulation "can expressed" should be replaced by "can be expressed"

Line 124: The word "someone" should be replaced "as shown"

Line 143: The formulation "which the third column" should be replaced with "which is the third column"

Line 157" The formulation "shall be as close to the" should be replaced by "shall be as close as possible to the"

Line 160: The formulation "shall be as similar to" should be replaced by "Shall be as similar as possible to"

Line 193: the formulation "closer to building" should be replaced by "closer to the building"

Line 205: The formulation "cell-like" should be replaced by "cell-like dendrite"

Line 231: The formulation "This indicates the solidified" should be replaced by "This indicates that the solidified"

Lines 250 and 252: The word "speed" should be replaced by "velocity"

The authors are suggested to perform a thorough check of the English style of the paper before submitting the final proof to the editor.

Author Response

Dear reviewer,

On behalf of my co-authors, we thank you very much for giving us an opportunity to revise our manuscript. We appreciate the reviewer very much for his/her careful revision and constructive comments on our manuscript.

We have studies reviewers’ comments carefully, and have made revision which is marked in red in the reviewed file. Replies to the comments point-by-point are as follows.

Thank you and best regards.

Sincerely yours,

Zhijian Shen

Comment 1: Line 28: The formulation “is one kind of metal additive manufacturing techniques” should be replaced by “is a metal additive manufacturing technology”

Response: The formulation “is one kind of metal additive manufacturing techniques” has been replaced by “is a metal additive manufacturing technology”

Comment 2: Lines 34 and 35: the formulation “3D structure of the substructure” should be replaced by something that avoids the repetition, for example “3D morphology of the substructure”. The formulation “3D structure of the substructure” is frequently used in the paper.

Response: All the “3D structure” have been replaced by “3D morphology”

Comment 3: Line 75 – the word “facility” should be replaced by “system”

Response: The word “facility” has been replaced by “system”.

Comment 4: Line 85, 89 and 91 – the word “about” should be removed.

Response: The word “about” has been removed.

Comment 5: Line 81, 82 and 254 – the term “power” should be replaced by “laser power”

Response: The term “power” has been replaced by “laser power”.

Comment 6: Line 121: the formulation “can expressed” should be replaced by “can be expressed”

Response: The formulation “can expressed” has been replaced by “can be expressed”.

Comment 7: Line 124: The word “someone” should be replaced “as shown”

Response: The word “someone” has been replaced by “as shown”.

Comment 8: Line 143: The formulation “which the third column” should be replaced with “which is the third column”

Response: The formulation “which the third column” has been replaced with “which is the third column”.

Comment 9: Line 157: The formulation “shall be as close to” should be replaced by “shall be as close as possible to”

Response: The formulation “shall be as close to” has been replaced by “shall be as close as possible to”.

Comment 10: Line 160: The formulation “shall be as similar to” should be replaced by “shall be as similar as possible to”

Response: The formulation “shall be as similar to” has been replaced by “shall be as similar as possible to”.

Comment 11: Line 205: The formulation “cell-like” should be replaced by “cell-like dendrite”

Response: We sinerely thank the reviewer for his/her suggestion. In this sentence, the word “cell-like” means that the morphologies of the substructure do not show any branching. We have not made the distinguish between cell and cell-like dendrite here. “As seen in the subplot in the PFs, the morphologies of the substructure do not show any branching, and they are cell-like.”

Comment 12: Line 231: the formulation “This indicates the solidified” should be replaced by “This indicates that the solidified”

Response: the formulation “This indicates the solidified” has been replaced by “This indicates that the solidified”.

Comment 13: Line 250 and 252: The word “speed” should be replaced by “velocity”

Response: The word “speed” has been replaced by “velocity”.

Comment 14: The authors are suggested to perform a thorough check of English style of the paper before submitting the final proof to the editor.

Response: We sincerely thank the reviewer for his/her valuable suggestion. We have performed a thorough check of English style of the paper.

Round 2

Reviewer 2 Report

The authors have addressed all issue raised previously and the manuscript is suitable for publication in its current form.

Author Response

We sincerely thank the reviewer for his/her positive comment.